# Variance Reduction for
# Stochastic Gradient Optimization

**Chong Wang    Xi Chen**[*]    **Alex Smola    Eric P. Xing**
Carnegie Mellon University,    University of California, Berkeley[*]
{chongw,xichen,epxing}@cs.cmu.edu    alex@smola.org

## Abstract

Stochastic gradient optimization is a class of widely used algorithms for training machine learning models. To optimize an objective, it uses the noisy gradient computed from the random data samples instead of the true gradient computed from the entire dataset. However, when the variance of the noisy gradient is large, the algorithm might spend much time bouncing around, leading to slower convergence and worse performance. In this paper, we develop a general approach of using *control variate* for variance reduction in stochastic gradient. Data statistics such as low-order moments (pre-computed or estimated online) is used to form the control variate. We demonstrate how to construct the control variate for two practical problems using stochastic gradient optimization. One is convex—the MAP estimation for logistic regression, and the other is non-convex—stochastic variational inference for latent Dirichlet allocation. On both problems, our approach shows faster convergence and better performance than the classical approach.

## 1   Introduction

Stochastic gradient (SG) optimization [1, 2] is widely used for training machine learning models with very large-scale datasets. It uses the noisy gradient (a.k.a. stochastic gradient) estimated from random data samples rather than that from the entire data. Thus, stochastic gradient algorithms can run many more iterations in a limited time budget. However, if the noisy gradient has a large variance, the stochastic gradient algorithm might spend much time bouncing around, leading to slower convergence and worse performance. Taking a mini-batch with a larger size for computing the noisy gradient could help to reduce its variance; but if the mini-batch size is too large, it can undermine the advantage in efficiency of stochastic gradient optimization.

In this paper, we propose a general remedy to the "noisy gradient" problem ubiquitous to all stochastic gradient optimization algorithms for different models. Our approach builds on a *variance reduction* technique, which makes use of control variates [3] to augment the noisy gradient and thereby reduce its variance. The augmented "stochastic gradient" can be shown to remain an unbiased estimate of the true gradient, a necessary condition that ensures the convergence. For such control variates to be effective and sound, they must satisfy the following key requirements: 1) they have a high correlation with the noisy gradient, and 2) their expectation (with respect to random data samples) is inexpensive to compute. We show that such control variates can be constructed via low-order approximations to the noisy gradient so that their expectation only depends on low-order moments of the data. The intuition is that these low-order moments roughly characterize the empirical data distribution, and can be used to form the control variate to correct the noisy gradient to a better direction. In other words, the variance of the augmented "stochastic gradient" becomes smaller as it is derived with more information about the data.

The rest of the paper is organized as follows. In §2, we describe the general formulation and the theoretical property of variance reduction via control variates in stochastic gradient optimization.

In §3, we present two examples to show how one can construct control variates for practical algorithms. (More examples are provided in the supplementary material.) These include a convex problem—the MAP estimation for logistic regression, and a non-convex problem—stochastic variational inference for latent Dirichlet allocation [22]. Finally, we demonstrate the empirical performance of our algorithms under these two examples in §4. We conclude with a discussion on some future work.

## 2 Variance reduction for general stochastic gradient optimization

We begin with a description of the general formulation of variance reduction via control variate for stochastic gradient optimization. Consider a general optimization problem over a finite set of training data $\mathcal{D} = \{x_d\}_{d=1}^D$ with each $x_d \in \mathbb{R}^p$. Here $D$ is the number of the training data. We want to maximize the following function with respect to a $p$-dimensional vector $w$,

$$\underset{w}{\text{maximize }} \mathcal{L}(w) := \mathcal{R}(w) + (1/D) \sum_{d=1}^D f(w; x_d),$$

where $\mathcal{R}(w)$ is a regularization function.[1] Gradient-based algorithms can be used to maximize $\mathcal{L}(w)$ at the expense of computing the gradient over the entire training set. Instead, stochastic gradient (SG) methods use the noisy gradient estimated from random data samples. Suppose data index $d$ is selected uniformly from $\{1, \cdots, D\}$ at step $t$,

$$g(w; x_d) = \nabla_w \mathcal{R}(w) + \nabla_w f(w; x_d), \tag{1}$$
$$w_{t+1} = w_t + \rho_t g(w; x_d), \tag{2}$$

where $g(w; x_d)$ is the noisy gradient that only depends on $x_d$ and $\rho_t$ is a proper step size. To make notation simple, we use $g_d(w) \triangleq g(w; x_d)$.

Following the standard stochastic optimization literature [1, 4], we require the expectation of the noisy gradient $g_d$ equals to the true gradient,

$$\mathbb{E}_d[g_d(w)] = \nabla_w \mathcal{L}(w), \tag{3}$$

to ensure the convergence of the stochastic gradient algorithm. When the variance of $g_d(w)$ is large, the algorithm could suffer from slow convergence.

The basic idea of using control variates for variance reduction is to construct a new random vector that has the same expectation as the target expectation but with smaller variance. In previous work [5], control variates were used to improve the estimate of the intractable integral in variational Bayesian inference which was then used to compute the gradient of the variational lower bound. In our context, we employ a random vector $h_d(w)$ of length $p$ to reduce the variance of the sampled gradient,

$$\widetilde{g}_d(w) = g_d(w) - A^T(h_d(w) - h(w)), \tag{4}$$

where $A$ is a $p \times p$ matrix and $h(w) \triangleq \mathbb{E}_d[h_d(w)]$. (We will show how to choose $h_d(w)$ later, but it usually depends on the form of $g_d(w)$.) The random vector $\widetilde{g}_d(w)$ has the same expectation as the noisy gradient $g_d(w)$ in Eq. 1, and thus can be used to replace $g_d(w)$ in the SG update in Eq. 2. To reduce the variance of the noisy gradient, the trace of the covariance matrix of $\widetilde{g}_d(w)$,

$$\text{Var}_d[\widetilde{g}_d(w)] \triangleq \text{Cov}_d[\widetilde{g}_d(w), \widetilde{g}_d(w)] = \text{Var}_d[g_d(w)]$$
$$- (\text{Cov}_d[h_d(w), g_d(w)] + \text{Cov}_d[g_d(w), h_d(w)])A + A^T \text{Var}_d[h_d(w)]A, \tag{5}$$

must be necessarily small; therefore we set $A$ to be the minimizer of $\text{Tr}\left(\text{Var}_d[\widetilde{g}_d(w)]\right)$. That is,

$$A^* = \text{argmin}_A \text{Tr}\left(\text{Var}_d[\widetilde{g}_d(w)]\right)$$
$$= (\text{Var}_d[h_d(w)])^{-1}\left(\text{Cov}_d[g_d(w), h_d(w)] + \text{Cov}_d[h_d(w), g_d(w)]\right)/2. \tag{6}$$

The optimal $A^*$ is a function of $w$.

**Why is $\widetilde{g}_d(w)$ a better choice?** Now we show that $\widetilde{g}_d(w)$ is a better "stochastic gradient" under the $\ell_2$-norm. In the first-order stochastic oracle model, we normally assume that there exists a constant $\sigma$ such that for any estimate $w$ in its domain [6, 7]:

$$\mathbb{E}_d\left[\|g_d(w) - \mathbb{E}_d[g_d(w)]\|_2^2\right] = \text{Tr}(\text{Var}_d[g_d(w)]) \le \sigma^2.$$

Under this assumption, the dominating term in the optimal convergence rate is $O(\sigma/\sqrt{t})$ for convex problems and $O(\sigma^2/(\mu t))$ for strongly convex problems, where $\mu$ is the strong convexity parameter (see the definition of strong convexity on Page 459 in [8]).

Now suppose that we can find a random vector $h_d(w)$ and compute $A^*$ according to Eq. 6. By plugging $A^*$ back into Eq. 5,

$$\mathbb{E}_d\left[\|\widetilde{g}_d(w) - \mathbb{E}_d[\widetilde{g}_d(w)]\|_2^2\right] = \mathrm{Tr}(\mathrm{Var}_d[\widetilde{g}_d(w)]),$$

where $\mathrm{Var}_d[\widetilde{g}_d(w)] = \mathrm{Var}_d[g_d(w)] - \mathrm{Cov}_d[g_d(w), h_d(w)](\mathrm{Var}_d[h_d(w)])^{-1}\mathrm{Cov}_d[h_d(w), g_d(w)]$.

For any estimate $w$, $\mathrm{Cov}_d(g_d, h_d)\left(\mathrm{Cov}_d(h_d, h_d)\right)^{-1}\mathrm{Cov}_d(h_d, g_d)$ is a semi-positive definite matrix. Therefore, its trace, which equals to the sum of the eigenvalues, is positive (or zero when $h_d$ and $g_d$ are uncorrelated) and hence,

$$\mathbb{E}_d\left[\|\tilde{g}_d(w) - \mathbb{E}_d[\tilde{g}_d(w)]\|_2^2\right] \leq \mathbb{E}_d\left[\|g_d(w) - \mathbb{E}_d[g_d(w)]\|_2^2\right].$$

In other words, it is possible to find a constant $\tau \leq \sigma$ such that $\mathbb{E}_d\left[\|\tilde{g}_d(w) - \mathbb{E}_d[\tilde{g}_d(w)]\|_2^2\right] \leq \tau^2$ for all $w$. Therefore, when applying stochastic gradient methods, we could improve the optimal convergence rate from $O(\sigma/\sqrt{t})$ to $O(\tau/\sqrt{t})$ for convex problems; and from $O(\sigma^2/(\mu t))$ to $O(\tau^2/(\mu t))$ for strongly convex problems.

**Estimating optimal $A^*$.** When estimating $A^*$ according to Eq. 6, one needs to compute the inverse of $\mathrm{Var}_d[h_d(w)]$, which could be computationally expensive. In practice, we could constrain $A$ to be a diagonal matrix. According to Eq. 5, when $A = \mathrm{Diag}(a_{11}, \ldots, a_{pp})$, its optimal value is:

$$a_{ii}^* = \frac{[\mathrm{Cov}_d(g_d(w), h_d(w))]_{ii}}{[\mathrm{Var}_d(h_d(w))]_{ii}}. \tag{7}$$

This formulation avoids the computation of the matrix inverse, and leads to significant reduction of computational cost since only the diagonal elements of $\mathrm{Cov}_d(g_d(w), h_d(w))$ and $\mathrm{Var}_d(h_d(w))$, instead of the full matrices, need to be evaluated. It can be shown that, this simpler surrogate to the $A^*$ due to Eq. 6 still leads to a better convergence rate. Specifically:

$$\mathbb{E}_d\left[\|\tilde{g}_d(w) - \mathbb{E}_d[\tilde{g}_d(w)]\|_2^2\right] = \mathrm{Tr}(\mathrm{Var}_d(\tilde{g}_d(w))) = \mathrm{Tr}\left(\mathrm{Var}_d(g_d(w))\right) - \sum_{i=1}^{p} \frac{([\mathrm{Cov}_d(g_d(w), h_d(w))]_{ii})^2}{[\mathrm{Var}_d(h_d(w))]_{ii}},$$
$$= \sum_{i=1}^{p}(1 - \rho_{ii}^2)\mathrm{Var}(g_d(w))_{ii} \leq \mathrm{Tr}\left(\mathrm{Var}_d(g_d(w))\right) = \mathbb{E}_d\left[\|g_d(w) - \mathbb{E}_d[g_d(w)]\|_2^2\right], \tag{8}$$

where $\rho_{ii}$ is the Pearson's correlation coefficient between $[g_d(w)]_i$ and $[h_d(w)]_i$.

Indeed, an even simpler surrogate to the $A^*$, by reducing $A$ to a single real number $a$, can also improve convergence rate of SG. In this case, according to Eq. 5, the optimal $a^*$ is simply:

$$a^* = \mathrm{Tr}\left(\mathrm{Cov}_d(g_d(w), h_d(w))\right)/\mathrm{Tr}\left(\mathrm{Var}_d(h_d(w))\right). \tag{9}$$

To estimate the optimal $A^*$ or its surrogates, we need to evaluate $\mathrm{Cov}_d(g_d(w), h_d(w))$ and $\mathrm{Var}_d(h_d(w))$ (or their diagonal elements), which can be approximated by the sample covariance and variance from mini-batch samples while running the stochastic gradient algorithm. If we can not always obtain mini-batch samples, we may use strategies like moving average across iterations, as those used in [9, 10].

From Eq. 8, we observe that when the Pearson's correlation coefficient between $g_d(w)$ and $h_d(w)$ is higher, the control variate $h_d(w)$ will lead to a more significant level of variance reduction and hence faster convergence. In the maximal correlation case, one could set $h_d(w) = g_d(w)$ to obtain zero variance. But obviously, we cannot compute $\mathbb{E}_d[h_d(w)]$ efficiently in this case. In practice, one should construct $h_d(w)$ such that it is highly correlated with $g_d(w)$. In next section, we will show how to construct control variates for both convex and non-convex problems.

## 3 Practicing variance reduction on convex and non-convex problems

In this section, we apply the variance reduction technique presented above to two exemplary but practical problems: MAP estimation for logistic regression—a convex problem; and stochastic variational inference for latent Dirichlet allocation [11, 22]—a non-convex problem. In the supplement,

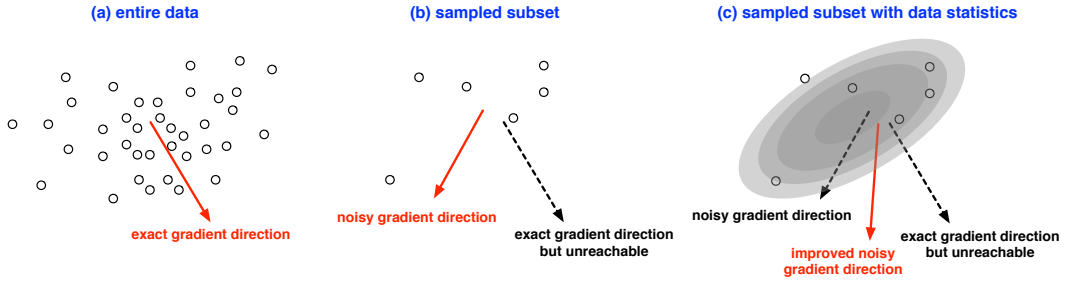

Figure 1: The illustration of how data statistics help reduce variance for the noisy gradient in stochastic optimization. The solid (red) line is the final gradient direction the algorithm will follow. (a) The exact gradient direction computed using the entire dataset. (b) The noisy gradient direction computed from the sampled subset, which can have high variance. (c) The improved noisy gradient direction with data statistics, such as low-order moments of the entire data. These low-order moments roughly characterize the data distribution, and are used to form the control variate to aid the noisy gradient.

we show that the same principle can be applied to more problems, such as hierarchical Dirichlet process [12, 13] and nonnegative matrix factorization [14].

As we discussed in §2, the higher the correlation between $g_d(w)$ and $h_d(w)$, the lower the variance is. Therefore, to apply the variance reduction technique in practice, the key is to construct a random vector $h_d(w)$ such that it has high correlations with $g_d(w)$, but its expectation $h(w) = \mathbb{E}_d[h_d(w)]$ is inexpensive to compute. The principle behind our choice of $h(w)$ is that we construct $h(w)$ based on some data statistics, such as low-order moments. These low-order moments roughly characterize the data distribution which does not depend on parameter $w$. Thus they can be pre-computed when processing the data or estimated online while running the stochastic gradient algorithm. Figure 1 illustrates this idea. We will use this principle throughout the paper to construct control variates for variance reduction under different scenarios.

## 3.1  SG with variance reduction for logistic regression

Logistic regression is widely used for classification [15]. Given a set of training examples $(x_d, y_d)$, $d = 1, ..., D$, where $y_d = 1$ or $y_d = -1$ indicates class labels, the probability of $y_d$ is

$$p(y_d \,|\, x_d, w) = \sigma(y_d w^\top x_d),$$

where $\sigma(z) = 1/(1 + \exp(-z))$ is the logistic function. The averaged log likelihood of the training data is

$$\ell(w) = \tfrac{1}{D} \sum_{d=1}^{D} \left\{ y_d w^\top x_d - \log\left(1 + \exp(y_d w^\top x_d)\right) \right\}. \tag{10}$$

An SG algorithm employs the following noisy gradient:

$$g_d(w) = y_d x_d \sigma(-y_d w^\top x_d). \tag{11}$$

Now we show how to construct our control variate for logistic regression. We begin with the first-order Taylor expansion around $\hat{z}$ for the sigmoid function,

$$\sigma(z) \approx \sigma(\hat{z})\left(1 + \sigma(-\hat{z})(z - \hat{z})\right).$$

We then apply this approximation to $\sigma(-y_d w^\top x_d)$ in Eq. 11 to obtain our control variate.[2] For logistic regression, we consider two classes separately, since data samples within each class are more likely to be similar. We consider positive data samples first. Let $z = -w^\top x_d$, and we define our control variate $h_d(w)$ for $y_d = 1$ as

$$h_d^{(1)}(w) \triangleq x_d \sigma(\hat{z})\left(1 + \sigma(-\hat{z})(z - \hat{z})\right) = x_d \sigma(\hat{z})\left(1 + \sigma(-\hat{z})(-w^\top x_d - \hat{z})\right).$$

Its expectation given $y_d = 1$ can be computed in closed-form as

$$\mathbb{E}_d[h_d^{(1)}(w) \,|\, y_d = 1] = \sigma(\hat{z})\left(\bar{x}^{(1)}\left(1 - \sigma(-\hat{z})\hat{z}\right) - \sigma(-\hat{z})\left(\mathrm{Var}^{(1)}[x_d] + \bar{x}^{(1)}(\bar{x}^{(1)})^\top\right)w\right),$$

where $\bar{x}^{(1)}$ and $\mathrm{Var}^{(1)}[x_d]$ are the mean and variance of the input features for the positive examples. In our experiments, we choose $\hat{z} = -w^\top \bar{x}^{(1)}$, which is the center of the positive examples. We can similarly derive the control variate $h_d^{(-1)}(w)$ for negative examples and we omit the details. Given the random sample regardless its label, the expectation of the control variate is computed as

$$\mathbb{E}_d[h_d(w)] = (D^{(1)}/D)\mathbb{E}_d[h_d^{(1)}(w)\,|\,y_d = 1] + (D^{(-1)}/D)\mathbb{E}_d[h_d^{(-1)}(w)\,|\,y_d = -1],$$

where $D^{(1)}$ and $D^{(-1)}$ are the number of positive and negative examples and $D^{(1)}/D$ is the probability of choosing a positive example from the training set. With Taylor approximation, we would expect our control variate is highly correlated with the noisy gradient. See our experiments in §4 for details.

### 3.2 SVI with variance reduction for latent Dirichlet allocation

The stochastic variational inference (SVI) algorithm used for latent Dirichlet allocation (LDA) [22] is also a form of stochastic gradient optimization, therefore it can also benefit from variance reduction. The basic idea is to stochastically optimize the variational objective for LDA, using stochastic mean field updates augmented by control variates derived from low-order moments on the data.

**Latent Dirichlet allocation (LDA).** LDA is the simplest topic model for discrete data such as text collections [17, 18]. Assume there are $K$ topics. The generative process of LDA is as follows.

1. Draw topics $\beta_k \sim \mathrm{Dir}_V(\eta)$ for $k \in \{1, \dots, K\}$.
2. For each document $d \in \{1, \dots, D\}$:
    (a) Draw topic proportions $\theta_d \sim \mathrm{Dir}_K(\alpha)$.
    (b) For each word $w_{dn} \in \{1, \dots, N\}$:
        i. Draw topic assignment $z_{dn} \sim \mathrm{Mult}(\theta_d)$.
        ii. Draw word $w_{dn} \sim \mathrm{Mult}(\beta_{z_{dn}})$.

Given the observed words $w \triangleq w_{1:D}$, we want to estimate the posterior distribution of the latent variables, including topics $\beta \triangleq \beta_{1:K}$, topic proportions $\theta \triangleq \theta_{1:D}$ and topic assignments $z \triangleq z_{1:D}$,

$$p(\beta, \theta, z\,|\,w) \propto \prod_{k=1}^K p(\beta_k\,|\,\eta) \prod_{d=1}^D p(\theta_d\,|\,\alpha) \prod_{n=1}^N p(z_{dn}\,|\,\theta_d)p(w_{dn}\,|\,\beta_{z_{dn}}). \tag{12}$$

However, this posterior is intractable. We must resort to approximation methods. Mean-field variational inference is a popular approach for the approximation [19].

**Mean-field variational inference for LDA.** Mean-field variational inference posits a family of distributions (called variational distributions) indexed by free variational parameters and then optimizes these parameters to minimize the KL divergence between the variational distribution and the true posterior. For LDA, the variational distribution is

$$q(\beta, \theta, z) = \prod_{k=1}^K q(\beta_k\,|\,\lambda_k) \prod_{d=1}^D q(\theta_d\,|\,\gamma_d) \prod_{n=1}^N q(z_{dn}\,|\,\phi_{dn}), \tag{13}$$

where the variational parameters are $\lambda_k$ (Dirichlet), $\theta_d$ (Dirichlet), and $\phi_{dn}$ (multinomial). We seek the variational distribution (Eq. 13) that minimizes the KL divergence to the true posterior (Eq. 12). This is equivalent to maximizing the lower bound of the log marginal likelihood of the data,

$$\log p(w) \geq \mathbb{E}_q\left[\log p(\beta, \theta, z, w)\right] - \mathbb{E}_q\left[\log q(\beta, \theta, z)\right] \triangleq \mathcal{L}(q), \tag{14}$$

where $\mathbb{E}_q[\cdot]$ denotes the expectation with respect to the variational distribution $q(\beta, \theta, z)$. Setting the gradient of the lower bound $\mathcal{L}(q)$ with respect to the variational parameters to zero gives the following coordinate ascent algorithm [17]. For each document $d \in \{1, \dots, D\}$, we run local variational inference using the following updates until convergence,

$$\phi_{dv}^k \propto \exp\left\{\Psi(\gamma_{dk}) + \Psi(\lambda_{k,v}) - \Psi\left(\sum_v \lambda_{kv}\right)\right\} \quad \text{for } v \in \{1, \dots, V\} \tag{15}$$

$$\gamma_d = \alpha + \sum_{v=1}^V n_{dv}\phi_{dv}. \tag{16}$$

where $\Psi(\cdot)$ is the digamma function and $n_{dv}$ is the number of term $v$ in document $d$. Note that here we use $\phi_{dv}$ instead of $\phi_{dn}$ in Eq. 13 since the same term $v$ have the same $\phi_{dn}$. After finding the variational parameters for each document, we update the variational Dirichlet for each topic,

$$\lambda_{kv} = \eta + \sum_{d=1}^D n_{dv}\phi_{dv}^k. \tag{17}$$

The whole coordinate ascent variational algorithm iterates over Eq. 15, 16 and 17 until convergence. However, this also reveals the drawback of this algorithm—updating the topic parameter $\lambda$ in Eq. 17 depends on the variational parameters $\phi$ from every document. This is especially inefficient for large-scale datasets. Stochastic variational inference solves this problem using stochastic optimization.

**Stochastic variational inference (SVI).** Instead of using the coordinate ascent algorithm, SVI optimizes the variational lower bound $\mathcal{L}(q)$ using stochastic optimization [22]. It draws random samples from the corpus and use these samples to form the noisy estimate of the natural gradient [20]. Then the algorithm follows that noisy natural gradient with a decreasing step size until convergence. The noisy gradient only depends on the sampled data and it is inexpensive to compute. This leads to a much faster algorithm than the traditional coordinate ascent variational inference algorithm.

Let $d$ be a random document index, $d \sim \text{Unif}(1, ..., D)$ and $\mathcal{L}_d(q)$ be the *sampled* lower bound. The sampled lower bound $\mathcal{L}_d(q)$ has the same form as the $\mathcal{L}(q)$ in Eq. 14 except that the sampled lower bound uses a virtual corpus that only contains document $d$ replicated $D$ times. According to [22], for LDA the noisy natural gradient with respect to the topic variational parameters is

$$g_d(\lambda_{kv}) \triangleq -\lambda_{kv} + \eta + Dn_{dv}\phi_{dv}^k, \tag{18}$$

where the $\phi_{dv}^k$ are obtained from the local variational inference by iterating over Eq. 15 and 16 until convergence.[3] With a step size $\rho_t$, SVI uses the following update $\lambda_{kv} \leftarrow \lambda_{kv} + \rho_t g_d(\lambda_{kv})$. However, the sampled natural gradient $g_d(\lambda_{kv})$ in Eq. 18 might have a large variance when the number of documents is large. This could lead to slow convergence or a poor local mode.

**Control variate.** Now we show how to construct control variates for the noisy gradient to reduce its variance. According to Eq. 18, the noisy gradient $g_d(\lambda_{kv})$ is a function of topic assignment parameters $\phi_{dv}$, which in turn depends on $w_d$, the words in document $d$, through the iterative updates in Eq. 15 and 16. This is different from the case in Eq. 11. In logistic regression, the gradient is an analytical function of the training data (Eq. 11), while in LDA, the natural gradient directly depends on the optimal local variational parameters (Eq. 18), which then depends on the training data through the local variational inference (Eq. 15). However, by carefully exploring the structure of the iterations, we can create effective control variates.

The key idea is to run Eq. 15 and 16 only up to a *fixed* number of iterations, together with some additional approximations to maintain analytical tractability. Starting the iteration with $\gamma_{dk}$ having the same value, we have $\phi_v^{k(0)} \propto \exp\left\{\Psi(\lambda_{kv}) - \Psi\left(\sum_v \lambda_{kv}\right)\right\}$.[4] Note that $\phi_v^{k(0)}$ does *not* depend on document $d$. Intuitively, $\phi_v^{k(0)}$ is the probability of term $v$ belonging to topic $k$ out of $K$ topics.

Next we use $\gamma_{dk} - \alpha$ to approximate $\exp(\Psi(\gamma_{dk}))$ in Eq. 15.[5] Plugging this approximation into Eq. 15 and 16 leads to the update,

$$\phi_{dv}^{k(1)} = \frac{\left(\sum_{u=1}^V f_{du}\phi_u^{k(0)}\right)\phi_v^{k(0)}}{\sum_{k=1}^K \left(\sum_{u=1}^V f_{du}\phi_u^{k(0)}\right)\phi_v^{k(0)}} \approx \frac{\left(\sum_{u=1}^V f_{du}\phi_u^{k(0)}\right)\phi_v^{k(0)}}{\sum_{k=1}^K \left(\sum_{u=1}^V \bar{f}_u\phi_u^{k(0)}\right)\phi_v^{k(0)}}, \tag{19}$$

where $f_{dv} = n_{dv}/n_d$ is the empirical frequency of term $v$ in document $d$. In addition, we replace $f_{du}$ with $\bar{f}_u \triangleq (1/D)\sum_d f_{du}$, the averaged frequency of term $u$ in the corpus, making the denominator of Eq. 19, $m_v^{(1)} \triangleq \sum_{k=1}^K \left(\sum_{u=1}^V \bar{f}_u\phi_u^{k(0)}\right)\phi_v^{k(0)}$, independent of documents. This approximation does not change the relative importance for the topics from term $v$. We define our control variate as

$$h_d(\lambda_{kv}) \triangleq Dn_{dv}\phi_{dv}^{k(1)},$$

whose expectation is $\mathbb{E}_d[h_d(\lambda_{kv})] = \left(D/m_v^{(1)}\right)\left\{\left(\sum_{u=1}^V \overline{n_v f_u}\phi_u^{k(0)}\right)\phi_v^{k(0)}\right\}$, where $\overline{n_v f_u} \triangleq (1/D)\sum_d n_{du}f_{dv} = (1/D)\sum_d n_{du}n_{dv}/n_d$. This depends on up to the second-order moments of data, which is usually sparse. We can continue to compute $\phi_{dv}^{k(2)}$ (or higher) given $\phi_{dv}^{k(1)}$, which turns out using the third-order (or higher) moments. We omit the details here. Similar ideas can be used in deriving control variates for hierarchical Dirichlet process [12, 13] and nonnegative matrix factorization [14]. We outline these in the supplementary material.

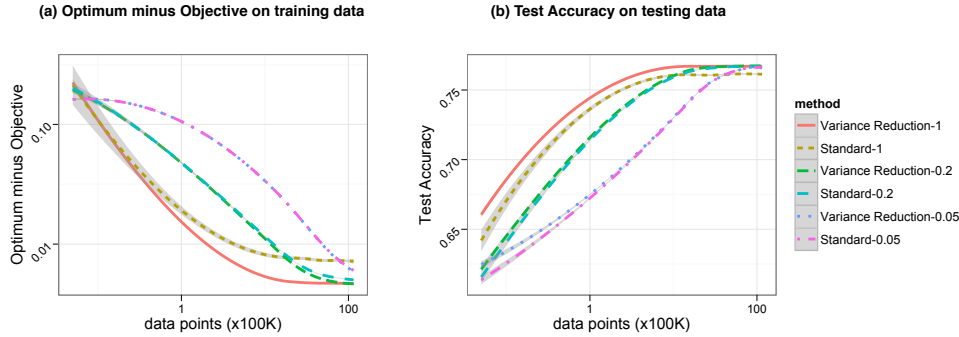

Figure 2: Comparison of our approach with standard SG algorithms using different constant learning rates. The figure was created using *geom_smooth* function in ggplot2 using local polynomial regression fitting (loess). A wider stripe indicates the result fluctuates more. This figure is best viewed in color. (Decayed learning rates we tested did not perform as well as constant ones and are not shown.) Legend "Variance Reduction-1" indicates the algorithm with variance reduction using learning rate $\rho_t = 1.0$. (a) Optimum minus the objective on the training data. The *lower* the better. (b) Test accuracy on testing data. The *higher* the better. From these results, we see that variance reduction with $\rho_t = 1.0$ performs the best, while the standard SG algorithm with $\rho_t = 1.0$ learns faster but bounces more (a wider stripe) and performs worse at the end. With $\rho_t = 0.05$, variance reduction performs about the same as the standard algorithm and both converge slowly. These indicate that with the variance reduction, a larger learning rate is possible to allow faster convergence without sacrificing performance.

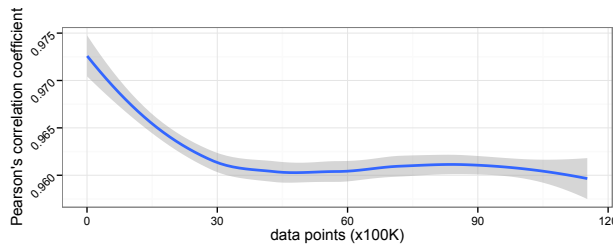

Figure 3: Pearson's correlation coefficient for $\rho_t = 1.0$ as we run the our algorithm. It is usually high, indicating the control variate is highly correlated with the noisy gradient, leading to a large variance reduction. Other settings are similar.

## 4 Experiments

In this section, we conducted experiments on the MAP estimation for logistic regression and stochastic variational inference for LDA.[6] In our experiments, we chose to estimate the optimal $a^*$ as a scalar shown in Eq. 9 for simplicity.

### 4.1 Logistic regression

We evaluate our algorithm on stochastic gradient (SG) for logistic regression. For the standard SG algorithm, we also evaluated the version with averaged output (ASG), although we did not find it outperforms the standard SG algorithm much. Our regularization added to Eq. 10 for the MAP estimation is $-\frac{1}{2D}w^\top w$. Our dataset contains *covtype* ($D = 581,012, p = 54$), obtained from the LIBSVM data website.[7] We separate 5K examples as the test set. We test two types of learning rates, constant and decayed. For constant rates, we explore $\rho_t \in \{0.01, 0.05, 0.1, 0.2, 0.5, 1\}$. For decayed rates, we explore $\rho_t \in \{t^{-1/2}, t^{-0.75}, t^{-1}\}$. We use a mini-batch size of 100.

**Results.** We found that the decayed learning rates we tested did not work well compared with the constant ones on this data. So we focus on the results using the constant rates. We plot three cases in Figure 2 for $\rho_t \in \{0.05, 0.2, 1\}$ to show the trend by comparing the objective function on the training data and the test accuracy on the testing data. (The best result for variance reduction is obtained when $\rho_t = 1.0$ and for standard SGD is when $\rho_t = 0.2$.) These contain the best results of

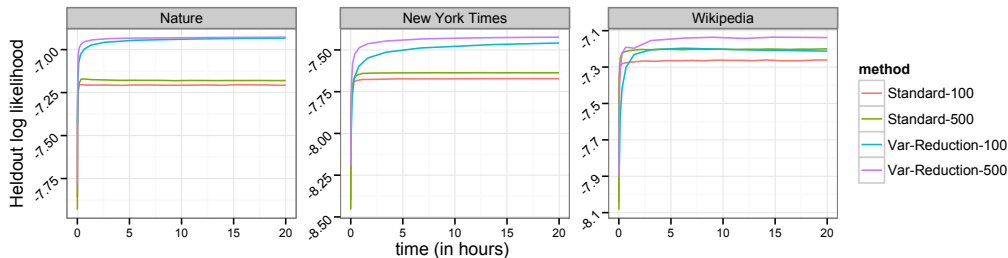

Figure 4: Held-out log likelihood on three large corpora. (Higher numbers are better.) Legend "Standard-100" indicates the stochastic algorithm in [10] with the batch size as 100. Our method consistently performs better than the standard stochastic variational inference. A large batch size tends to perform better.

each. With variance reduction, a large learning rate is possible to allow faster convergence without sacrificing performance. Figure 3 shows the mean of Pearson's correlation coefficient between the control variate and noisy gradient[8], which is quite high—the control variate is highly correlated with the noisy gradient, leading to a large variance reduction.

## 4.2 Stochastic variational inference for LDA

We evaluate our algorithm on stochastic variational inference for LDA. [10] has shown that the adaptive learning rate algorithm for SVI performed better than the manually tuned ones. So we use their algorithm to estimate adaptive learning rate. For LDA, we set the number of topics $K = 100$, hyperparameters $\alpha = 0.1$ and $\eta = 0.01$. We tested mini-batch sizes as 100 and 500.

**Data sets.** We analyzed three large corpora: *Nature*, *New York Times*, and *Wikipedia*. The *Nature* corpus contains 340K documents and a vocabulary of 4,500 terms; the *New York Times* corpus contains 1.8M documents and a vocabulary vocabulary of 8,000 terms; the *Wikipedia* corpus contains 3.6M documents and a vocabulary of 7,700 terms.

**Evaluation metric and results.** To evaluate our models, we held out 10K documents from each corpus and calculated its predictive likelihood. We follow the metric used in recent topic modeling literature [21, 22]. For a document $w_d$ in $\mathcal{D}_{\text{test}}$, we split it in into halves, $w_d = (w_{d1}, w_{d2})$, and computed the predictive log likelihood of the words in $w_{d2}$ conditioned on $w_{d1}$ and $\mathcal{D}_{\text{train}}$. The per-word predictive log likelihood is defined as

$$\text{likelihood}_{\text{pw}} \triangleq \sum_{d \in \mathcal{D}_{\text{test}}} \log p(w_{d2} | w_{d1}, \mathcal{D}_{\text{train}}) / \sum_{d \in \mathcal{D}_{\text{test}}} |w_{d2}|.$$

Here $|\cdot|$ is the number of words. A better predictive distribution given the first half gives higher likelihood to the second half. We used the same strategy as in [22] to approximate its computation. Figure 4 shows the results. On all three corpora, our algorithm gives better predictive distributions.

## 5 Discussions and future work

In this paper, we show that variance reduction with control variates can be used to improve stochastic gradient optimization. We further demonstrate its usage on convex and non-convex problems, showing improved performance on both. In future work, we would like to explore how to use second-order methods (such as Newton's method) or better line search algorithms to further improve the performance of stochastic optimization. This is because, for example, with variance reduction, second-order methods are able to capture the local curvature much better.

**Acknowledgement.** We thank anonymous reviewers for their helpful comments. We also thank Dani Yogatama for helping with some experiments on LDA. Chong Wang and Eric P. Xing are supported by NSF DBI-0546594 and NIH 1R01GM093156.

## Footnotes

[1] We follow the convention of maximizing a function $f$: when we mention a convex problem, we actually mean the objective function $-f$ is convex.

[2]Taylor expansion is not the only way to obtain control variates. Lower bounds or upper bounds of the objective function [16] can also provide alternatives. But we will not explore those solutions in this paper.

[3]Running to convergence is essential to ensure the natural gradient is valid in Eq. 18 [22].

[4]In our experiments, we set $\phi_v^{k(0)} = 0$ if $\phi_v^{k(0)}$ is less than 0.02. This leaves $\phi^{(0)}$ very sparse, since a term usually belongs to a small set of topics. For example, in Nature data, only 6% entries are non-zero.

[5]The scale of the approximation does not matter—$C(\gamma_{dk} - \alpha)$, where $C$ is a constant, has the same effect as $\gamma_{dk} - \alpha$. Other approximations to $\exp(\Psi(\gamma_{dk}))$ can also be used as long as it is linear in term of $\gamma_{dk}$.

[6]Code will be available on authors' websites.

[7]http://www.csie.ntu.edu.tw/~cjlin/libsvmtools/datasets

[8]Since the control variate and noisy gradient are vectors, we use the mean of the Pearson's coefficients computed for each dimension between these two vectors.

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
