[Supplementary Material]

# Variance Reduction for Stochastic Variational Inference

**Chong Wang    Xi Chen**[*]    **Alex Smola    Eric P. Xing**
Carnegie Mellon University,    University of California, Berkeley[*]
{chongw,xichen,epxing}@cs.cmu.edu    alex@smola.org

## 1  Applications to hierarchical Dirichlet process and nonnegative matrix factorization

### A. Hierarchical Dirichlet process topic models

Hierarchical Dirichlet process topic models uses two-level Dirichlet process [1]. The base distribution $H$ of the top-level DP is a symmetric Dirichlet over the vocabulary simplex—its atoms are topics. We draw once from this DP, $G_0 \sim \text{DP}(\omega, H)$. In the second level, we use $G_0$ as a base measure to a document-level DP, $G_d \sim \text{DP}(\alpha, G_0)$, which has the same set of atoms as $G_0$. We then draw the words of each document $d$ from topics from $G_d$. However, this representation is difficult for variational inference. Next, we review the stick-breaking construction representation for the (HDP) topic model used in [2, 3], which is convenient for developing variational inference algorithm.

The stick-breaking generative process of the HDP topic model is as follows.

1. Draw an infinite number of topics, $\beta_i \sim \text{Dir}(\eta)$ for $k = \{1, 2, 3, \ldots\}$.

2. Draw corpus breaking proportions, $v_k \sim \text{Beta}(1, \omega)$ for $k = \{1, 2, 3, \ldots\}$.

3. For each document $d$:

    (a) Draw document-level topic indices, $c_{di} \sim \text{Mult}(\sigma(v))$ for $i = \{1, 2, 3, \ldots\}$.
    (b) Draw document breaking proportions, $\pi_{di} \sim \text{Beta}(1, \alpha)$ for $i = \{1, 2, 3, \ldots\}$.
    (c) For each word $n$:
        i. Draw topic assignment $z_{dn} \sim \text{Mult}(\sigma(\pi_d))$.
        ii. Draw word $w_n \sim \text{Mult}(\beta_{c_{d,z_{dn}}})$.

Here the notation $\sigma(v)$ and $\sigma(\pi_d)$ are stick-breaking proportions defined as,

$$\sigma_i(v) = v_i \prod_{j=1}^{i-1}(1 - v_j),$$
$$\sigma_i(\pi_d) = \pi_{di} \prod_{j=1}^{i-1}(1 - \pi_{dj}).$$

As in [2, 3], we use a truncated variational family. At the corpus level, we truncate at $K$, fitting posteriors up to $K$ topics. At the document level we truncate at $T$, letting each document take $T$ topic indices. The variational distribution is as follows,

$$q(\beta, v, z, \pi) = \left( \prod_{k=1}^{K} q(\beta_k \mid \lambda_k) q(v_k \mid a_k) \right) \left( \prod_{d=1}^{D} \prod_{i=1}^{T} q(c_{di} \mid \zeta_{di}) q(\pi_{di} \mid \gamma_{di}) \prod_{n=1}^{N} q(z_{dn} \mid \phi_{dn}) \right),$$

where the variational parameters are $\lambda_k$ (Dirichlet, corpus-level), $a_k$ (Beta, corpus-level), $\xi_{di}$ (multinomial, document-level), $\gamma_{di}$ (Beta, document-level) and $\phi_{dn}$ (multinomial, word-level). We omit the detailed coordinate ascent updates for these parameters. Interested readers can refer to [3] for more information.

In stochastic variational inference, for a random sampled document $d$, we write down the noisy natural gradient for corpus-level variational parameters,

$$g_d(\lambda_{kv}) = -\lambda_{kv} + \eta + D \sum_{i=1}^{T} \zeta_{di}^k \sum_{n=1}^{N} \phi_{dn}^i I[w_{dn} = v],$$
$$g_d(a_k^{(1)}) = -a_k^{(1)} + 1 + D \sum_{i=1}^{T} \zeta_{di}^k,$$
$$g_d(a_k^{(2)}) = -a_k^{(2)} + \omega + D \sum_{i=1}^{T} \sum_{\ell=k+1}^{K} \zeta_{di}^\ell.$$

**Control variates.** For parameter $\lambda_{kv}$, $\sum_{i=1}^{T} \zeta_{di}^k \sum_{n=1}^{N} \phi_{dn}^i I[w_{dn} = v]$ gives the expected number of times that term $v$ in document $d$ is assigned to topic $k$ according to the variational distribution. This is similar to the natural gradient in Eq. 18 in the main paper used in LDA except that in HDP topic models, this is calculated via the topic indices $c_{di}$. So we choose the same control variates $\phi_v^k$ as that in LDA as in Eq. 19 in the main paper.

For parameter $a_k$, $\sum_{i=1}^{T} \zeta_{di}^k$ indicates the popularity of the topic $k$ in document $d$ by considering all topic indices. This could have a high correlation with the number of words assigned to topic $k$ in document $d$. Thus we control variates are

$$h_d(a_k^{(1)}) = D \sum_{v=1}^{V} \phi_v^k n_{dv},$$
$$h_d(a_k^{(2)}) = D \sum_{\ell=k+1}^{K} \sum_{v=1}^{V} \phi_v^\ell n_{dv}.$$

## B. Nonnegative matrix factorization

Now we show we can use the same idea for nonnegative matrix factorization (NMF) [4] given the connections between NMF, LDA and probabilistic semantic indexing [5, 6].

Suppose we have a non-negative dataset, $x = \{x_1, x_2, ..., x_D\}$, and each $x_d$ is a length-$V$ vector. We assume the factorization is obtained by minimizing

$$D(x||\beta\theta) \triangleq \sum_{d=1}^{D} D(x_d||\beta\theta_d),$$

where $\theta_{dk} \geq 0$, for $k = 1, ..., K$, where $K$ is the latent dimensions of NMF. Let $\beta = [\beta_1, ..., \beta_K]$ be the basis,

$$\sum_{v=1}^{V} \beta_{kv} = 1 \text{ and } \beta_{kv} \geq 0. \tag{1}$$

The distance metric is the generalized KL-divergence [4],

$$D(x_d||\beta\theta_d) = \sum_{v=1}^{V} \left( x_{dv} \log \frac{x_{dv}}{\sum_{k=1}^{K} \beta_{kv}\theta_{dk}} - x_{dv} + \sum_{k=1}^{K} \beta_{kv}\theta_{dk} \right)$$
$$= \sum_{v=1}^{V} \left( x_{dv} \left( \log x_{dv} - \log \sum_{k=1}^{K} \beta_{kv}\theta_{dk} \right) - x_{dv} \right) + \sum_{k=1}^{K} \theta_{dk}.$$

To minimize this metric, we choose to use an EM-style algorithm as follows. Let $\sum_{k=1}^{K} \phi_{dv}^k = 1$, then we can lower bound it using the Jensen's inequality

$$\log \sum_{k=1}^{K} \beta_{kv}\theta_{dk} = \log \sum_{k=1}^{K} \frac{\beta_{kv}\theta_{dk}}{\phi_{dv}^k} \phi_{dv}^k \geq \sum_{k=1}^{K} (\phi_{dv}^k \log \beta_{kv}\theta_{dk} - \log \phi_{dv}^k),$$

where the optimal $\phi_{dv}^k$ is

$$\phi_{dv}^k \propto \beta_{kv}\theta_{dk},$$

and this gives the *tight* bound. Then the update for $\theta_d$ and $\beta$ is

$$\theta_{dk} = \sum_{v=1}^{V} x_{dv}\phi_{dv}^k,$$
$$\beta_{kv} \propto \sum_{d=1}^{D} x_{dv}\phi_{dv}^k. \tag{2}$$

However, the update $\beta_{kv}$ does not allow us easily to use a natural gradient algorithm that is similar to LDA or HDP. We change the objective as follows. Assume

$$p(\beta \,|\, \eta) = \prod_k \mathrm{Dir}(\beta_k \,|\, \eta).$$

We will find $q(\beta) = \prod_k q(\beta_k \,|\, \lambda_k)$ that minimizes

$$\sum_{d=1}^{D} \mathrm{E}_q[D(x_d||\beta\theta_d)] + KL\left(q(\beta|\lambda)||p(\beta \,|\, \eta)\right).$$

Minimizing this leads to the updates

$$\phi_{dv}^k \propto \theta_{dk} \exp\left\{\Psi(\lambda_{k,v}) - \Psi\left(\sum_v \lambda_{kv}\right)\right\},$$
$$\theta_{dk} = \sum_{v=1}^{V} x_{dv} \phi_{dv}^k,$$

The natural gradient with respect to $\lambda_{kv}$ is

$$g_d(\lambda_{kv}) = -\lambda_{kv} + \eta + \sum_{d=1}^{D} x_{dv} \phi_{dv}^k. \tag{3}$$

Eq. 3 lets us use the variance reduction technique presented in the main paper.