[Reviews · NeurIPS 2013]

Submitted by Assigned_Reviewer_3

DETAILED COMMENTS:

This work considers using control variates to reduce the variations of stochastic gradient descent algorithms, a very important problem for all stochastic approximation methods. This approach is not new to the machine learning community, see e.g. "John Paisley, David M. Blei and Michael I. Jordan, Variational Bayesian Inference with Stochastic Search, in: Proceedings of the 29th International Conference on Machine Learning, Edinburgh, Scotland, UK, 2012", but the authors provide more examples for popular models such as LDA and NMF. The paper also includes some experimental results in the context of logistic regression and LDA.

The applications of control variates in logistic regression and variational inference were presented in [Paisley'2012], although they are not exactly the same as the ones used in this paper. The authors should at least cite this paper, elaborate the novelties of this work, and compare with the choices of control variates therein.

In the experiments for logistic regression, the comparisons between using variance reduction and standard sgd is based on the SAME fixed stepsize. This does not sound reasonable to me. A fair comparison should choose the best stepsizes separately for variance reduction and standard sgd.

What is the additional overhead for using control variates? Since there is no computational complexity analysis, comparisons using CPU time or wall-clock time are desired.
Summary: This work uses control variates to reduce the variations of stochastic gradient descent algorithms. The novelty of this paper has to be clarified and choice of fixed stepsizes for both variance reduction and standard sgd in logistic regression needs justification.

Submitted by Assigned_Reviewer_6

quality: 5 (out of 10)
clarity: 6
originality: 6
significance: 9

SUMMARY: The authors propose to accelerate the stochastic gradient optimization algorithm by reducing the variance of the noisy gradient estimate by using the 'control variate' trick (a standard variance reduction technique for Monte Carlo simulations, explained in [3] for example). The control variate is a vector which hopefully has high correlation with the noisy gradient but for which the expectation is easier to compute. Standard convergence rates for stochastic gradient optimization depend on the variance of the gradient estimates, and thus a variance reduction technique should yield an acceleration of convergence. The authors give examples of control variates by using Taylor approximations of the gradient estimate for the optimization problem arising in regularized logistic regression as well as for MAP estimation for the latent Dirichlet Allocation (LDA) model. They compare constant step-size SGD with and without variance reduction for logistic regression on the covtype dataset, claiming that the variance reduction allows to use bigger step-sizes without having the problem of high variance and thus yields faster empirical convergence. For LDA, they compare the adaptive step-size version of the stochastic optimization method of [10] with and without variance reduction, showing a faster convergence on the held-out test log-likelihood on three large corpora.

EVALUATION:
Pros:
- I like the general idea of variance reduction for SGD using control variates -- it could have a big impact given the popularity of SGD.
- The motivation is compelling; the concrete examples of control variates are convincing; and the the general idea (Taylor approximation to define them) seems generalizable
- The paper is fairly easy to read.

Cons:
- The experiments are somewhat weak: only one dataset for logistic regression; and a lack of standardized setup for LDA.
- The related work is not covered.

QUALITY: The theoretical motivation for the approach is compelling (reducing the variance of the gradient estimates reduces the constant in the convergence rate), but the execution in the empirical section is fairly weak.
1) For logistic regression, they only consider one dataset (covtype). As the previous SGD optimization literature has showed, there can be significant variations of behavior between different datasets (and step-sizes) -- see for example figure 1 and 2 of "A Stochastic Gradient Method with an Exponential Convergence Rate for Finite Training Sets", in the arXiv:1202.6258v4 version which compare SGD with different methods on covtype and other datasets for regularized logistic regression.
2) For the LDA experiments, I am puzzled why the authors didn't re-use exactly the same setup as in [10] and [4] so that their results could be comparable? Why using a different mini-batch size (500 vs. 100); a different held-out test set (2k vs. 10k), etc.? It is suspicious that the results of their baseline on the held-out test set (which is the state-of-the-art method in 10] are all systematically worse than what was presented in [10]. The authors should clarify this in their rebuttal.
3) Another compelling experimental baseline which is missing is to compare using different mini-batch sizes (which is also a variance reduction technique mentioned in the introduction) vs. their control variate method -- as shown in [4], the

CLARITY: The paper is fairly easy to read. I like figure 1. An important point which needs to be clarified is how do they estimate the covariance quantities in their experiments to compute a* (do they use the empirical covariances on the mini-batch? This should be repeated in the experiments section -- and perhaps a discussion of how its cost compare to the standard SGD method should be included). Figure 2 is very hard to read -- the authors should add *markers* to identify the different lines. See also below for more suggestions.

ORIGINALITY: I am not familiar of any work using such variance reduction techniques for SGD in such generality. On the other hand, the paper is lacking a coverage of related work. An important missing reference is the paper "Variational Bayesian Inference with Stochastic Search" by John Paisley, David Blei, Michael Jordan ICML 2012 which also uses a control variate method to reduce the variance of a stochastic optimization approach to variational mean-field to do variational Bayes (they consider both Bayesian logistic regression and HDPs). The authors should explain in their rebuttal what novel contributions they make in comparison to this prior work. Another relevant piece of work (which is a useful pointer to mention though it doesn't compete with the novelty of this submission) is "Variance Reduction Techniques for Gradient Estimates in Reinforcement Learning" by Evan Greensmith, Peter L. Bartlett, Jonathan Baxter, JMLR 2004.

SIGNIFICANCE: In addition to the two applications mentioned in this paper, the approach presented could be most probably generalized to many other settings where SGD has been used. Given the popularity of SGD for large-scale optimization, the potential impact of this work is quite significant. The empirical evidence presented is somewhat weak, but the theoretical intuition is fairly compelling and I could believe that a more thorough empirical comparison could also show significant improvements. I note that their theoretical argument didn't depend on having a finite training set; it would thus be interesting to see this approach used as well in the real stochastic optimization setting (where the full expectation cannot be computed) and where running averages are used to estimate the quantities.

== Other detailed comments ==

line 058: 'discuss' - > discussion

line 089: The authors should be clearer that the matrix A *depends on w* as well.

equation (5): Cov(g, h) is not symmetric in general -- so the 2nd term should be -(cov(g,h) + cov(g,h)^T).

equation (6): it should be cov(h,g) on the RHS [or cov(g,h)^T], not cov(g,h)

Paragraph 150-154: It might be worthwhile to point out that in the maximal correlation case, one could set h_d = g_d and then the variance becomes zero (but obviously, we cannot compute E[h_d] efficiently in this case).

lines 295-296: "This is different from the case in Eq. 11 for logistic regression, which is explicit." - > this sentence is ambiguous. What is explicit?


figure 3: For which w did they compute the Pearson's coefficient? Is this using the true covariances or the estimated covariances from the mini-batch?

=== Update after rebuttal ==

The authors should carefully cover the related work in their final version of the paper, as well as implement the other corrections mentioned above (I will double check!). I agree that they make a nice contribution over the work by [Paisley et al. 2012]; on the other hand, they should also be clear that [Paisely et al. 2012] were already using control variates to improve the *optimization* of their variational objective; just that perhaps they were not using it in such generality as in this submission. I still think that the experiments are on the weak side, which is why my recommendation for acceptance is not stronger.
Summary: I like the idea of variance reduction for SGD and the authors give compelling examples on logistic regression and LDA. This idea could have significant impact. On the other hand, the execution in the empirical section is fairly weak.

Submitted by Assigned_Reviewer_8

Variance Reduction for Stochastic Gradient Optimization

NOTE: Due to the number of papers to review and the short reviewing time, to maintain consistency I did not seek to verify the correctness of the results. As a result, I generally ignored any proofs or additional details in the supplementary material. This review should be considered as a review of the main paper alone.

This paper proposes a trick using control variates to help reduce the variance of the gradient step in stochastic gradient descent procedures. The idea is to use a structured correlation to reduce the variance, which is different than the averaging approach used in minibatching. The intuition is clear and the paper is generally well written. I believe that the contribution is somewhat novel (although the concept is not that new) and the audience at NIPS will appreciate it.

STRENGTHS:
- develops a trick to help improve the performance of SGD that is different from minibatching and applies it to specific problem
- the experiments show that this trick works, so this is an important tool to add to the arsenal

WEAKNESSES:
- it's hard to tell how one can construct control variates for other problems

COMMENTS: This is a rather nice implementation-driven paper which tries to address one of the big problems with SGD -- it is great on paper and rather finicky in practice. I found the presentation convincing, but some of the claims feel a bit overstated to me.

One issue is the concept of novelty. They do point out that control variates have been used elsewhere, but they are missing a reference to a recent work by Paisely et al (ICML 2012) which also uses control variates for variance reduction in gradient estimation and applies it to HDPs... while it's hard to keep on top of all of the literature, the novelty with respect to this previous approach is unclear.

Because many of the experimental details seemed a bit arbitrary (e.g. mini batch size of 100), the comparisons between this approach and minibatching felt overstated. The approach of "Algorithm A does better than Algorithm B on theses datasets" doesn't tell me *why*, especially, when it was just one run with a fixed set of learning rates. Will hand-tuning the learning rates help? Who knows? I think the authors should instead focus on exploring how much this trick can help and when/where there may be diminishing returns or interesting practical tradeoffs to explore.

"It can be shown that, this simpler surrogate to the A∗ due to Eq. 6 still leads to a better convergence rate." -- for this and other comments I would prefer that there be explicit references to the supplementary material (if a proof exists) or omitted (if it does not).

Does this work with proper gradients only, or can it be applied to subgradients as well?

TYPOS/SMALL ITEMS:
058: "discussion on"
before (4): be clear that $h$ can depend on $g$ here.

ADDITIONAL COMMENTS AFTER THE REBUTTAL:
* With regards to novelty, it's more about tone than substance -- the idea of using control variates to help reduce variance is not new (as the authors note). I agree that the current application is sufficiently different than the ICML paper referenced above.
* Since I didn't have as extensive comments, I am happy with the response and am modifying my score.
Summary: This paper proposes a trick using control variates to help reduce the variance of the gradient step in stochastic gradient descent procedures. I believe that the contribution is novel (although the approach is known in the control literature) and the audience at NIPS should appreciate it.
Author Feedback

Author rebuttal: We thank reviewers for helpful comments and suggestions. We've addressed your questions and concerns as below.

Regarding novelty of the paper: we first thank the reviewers for pointing out the related work, specifically the reference of Paisley et al. at ICML 2012. The major difference is that we are solving completely different problems---they use the control variate to improve the estimate of the intractable integral in variational inference whereas we use the control variate to improve the estimate the noisy gradient in general stochastic gradient optimization. This leads to totally different ways of using control variates. In Paisley et al., they need control variates whose expectations are in closed-form under the variational distribution; in contrast, our control variates need to depend on only the lower-moments of the training data and thus do not have such a requirement. We will add discussions to differentiate our work from Paisley et al. at ICML 2012 to the updated version.

***Assigned_Reviewer_3

1) About using the SAME step size for comparison. In figure 2, we actually did *not* only report the same step sizes for comparison. There are three different step sizes (out of six we tested) for each method, and they contain the best ones of each---the standard sgd is best when rho=0.2 while variance reduction is when rho=1.0.

We did a grid search for both fixed step sizes and decreasing step sizes and we found fixed step sizes worked better on this dataset. (see line 366-367) So in figure 2, we only report the fixed ones due to space constraints.

2) About the overhead. The added overhead is no more than 10%, which is relatively small compared with the gain.


***Assigned_Reviewer_6

1) Estimating the covariance quantities a*. It is based on the empirical estimate from mini-batches from Eq. 9 in line 143.

2) About LDA setting. In our submission, we found batch size 100 sometimes was not very stable (so we used 500) and later we found it was due to a different initialization after we consulted the code writer of [10]. We had rerun all experiments using new settings.

3) About the “explicit” in line 295­-296. We agree this is ambiguous. By “explicit” in logistic regression, we meant the gradient in logistic regression is an analytical function of the training data (Eq 11 in line 198), while in LDA, the natural gradient directly depends on the optimal local variational parameters (Eq 18 in line 285-286), which then depends on the training data through the local variational inference (Eq 15 and 16 in line 263-265) --- thus “implicit” on the data.

4) About the Pearson's coefficient in Figure 3. This is between the noisy gradient and control variate. Since they are both vectors, we use the mean of the Pearson's coefficients computed for each dimension between these two vectors.

***Assigned_Reviewer_8

1) About constructing control variates. Taylor expansion is a good way for convex optimization problems. For stochastic variational inference, it can be a bit challenging in general settings.

2) The fact that a simpler surrogate to the A^* (a real number a) leads to faster convergence is a special case of the previous settings (matrix A^* or diagonal matrix A^*). Therefore, the proof is omitted due to space constraints.

3) About subgradients. Our variance reduction technique is a general framework and can be applied to both proper gradient and subgradient. For both cases, it leads to faster convergence.